# Clinical Significance of Multiparametric Magnetic Resonance Imaging as a Preoperative Predictor of Oncologic Outcome in Very Low-Risk Prostate Cancer

**DOI:** 10.3390/jcm8040542

**Published:** 2019-04-19

**Authors:** Doo Yong Chung, Min Seok Kim, Jong Soo Lee, Hyeok Jun Goh, Dong Hoon Koh, Won Sik Jang, Chang Hee Hong, Young Deuk Choi

**Affiliations:** 1Department of Urology, Inha University School of Medicine, 366 Seohae-daero, Jung-gu, Incheon 22332, Korea; wjdendyd@gmail.com; 2Department of Urology, Urological Science Institute, Yonsei University College of Medicine, 50-1 Yonsei-ro, Seodaemun-gu, Seoul 03722, Korea; kmsvvvv@naver.com (M.S.K.); JS1129@yuhs.ac (J.S.L.); JOON8301@yuhs.ac (H.J.G.); SINDAKJANG@yuhs.ac (W.S.J.); CHHONG52@yuhs.ac (C.H.H.); 3Department of Urology, Konyang University College of Medicine, 158 Gwanjeodong-ro, Daejeon 35365, Korea; urodhkoh@kyuh.ac.kr

**Keywords:** multiparametric MRI, preoperative predictor, oncologic outcome, very low-risk prostate cancer

## Abstract

Currently, multiparametric magnetic resonance imaging (mpMRI) is not an indication for patients with very low-risk prostate cancer. In this study, we aimed to evaluate the usefulness of mpMRI as a diagnostic tool in these patients. We retrospectively analyzed the clinical and pathological data of individuals with very low-risk prostate cancer, according to the NCCN guidelines, who underwent mpMRI before radical prostatectomy at our institution between 2010 and 2016. Patients who did not undergo pre-evaluation with mpMRI were excluded. We analyzed the factors associated with biochemical recurrence (BCR) using Cox regression model, logistic regression analysis, and Kaplan–Meier curve. Of 253 very low-risk prostate cancer patients, we observed 26 (10.3%) with BCR during the follow-up period in this study. The median follow-up from radical prostatectomy was 53 months (IQR 33–74). The multivariate Cox regression analyses demonstrated that the only factor associated with BCR in very low-risk patients was increase in the pathologic Gleason score (GS) (HR: 2.185, *p*-value 0.048). In addition, multivariate logistic analyses identified prostate specific antigen (PSA) (OR: 1.353, *p*-value 0.010), PSA density (OR: 1.160, *p*-value 0.013), and suspicious lesion on mpMRI (OR: 1.995, *p*-value 0.019) as the independent preoperative predictors associated with the pathologic GS upgrade. In our study, the pathologic GS upgrade after radical prostatectomy in very low-risk prostate cancer patients demonstrated a negative impact on BCR and mpMRI is a good prognostic tool to predict the pathologic GS upgrade. We believe that the implementation of mpMRI would be beneficial to determine the treatment strategy for these patients.

## 1. Introduction

Prostate cancer (PCa) is the most common newly diagnosed malignancy in males and its incidence has increased over the past 20 years due to the widespread use of prostate specific antigen (PSA) screening [1]. This trend has increased the diagnosis of localized PCa but a significant decrease in metastatic PCa has been reported [2,3,4]. Various treatment strategies for such localized PCa currently exist, such as surgery, radiation, and active surveillance [5]. Radical prostatectomy (RP) is one such effective treatment strategy for localized PCa [6]. However, the side effects, such as incontinence and erectile dysfunction, which occur after surgery, may be present [7,8]. Therefore, several guidelines define low-risk PCa and recommend noninvasive treatment, such as active surveillance for these patients [9,10]. This is to minimize overtreatment in patients. Also, several guidelines such as NCCN guideline further subdivided very low risk PCa in low risk PCa [10]. Some studies have shown that adverse oncological outcomes, including biochemical recurrence (BCR) and adverse pathological features (pT3 or pathological Gleason score upgrade) can present in patients with very low-risk PCa [11,12]. Therefore, we believe that active treatment may be required selectively in some very low risk PCa patients who may have adverse oncologic outcomes. Currently, magnetic resonance imaging (mpMRI) has increasingly been used for PCa diagnosis because of its growing availability, multiparametric imaging, and the combination of anatomic and functional data [13]. Moreover, the development of the Prostate Imaging Reporting and Data System (PI-RADS) with standardized MRI reading for PCa has an important role in the diagnosis and treatment decisions of PCa [14,15]. However, mpMRI is not indicated in patients with low-risk PCa according to the NCCN guidelines. We believe that mpMRI is of diagnostic value in patients with very low-risk PCa. Thus, we aimed to investigate the oncologic outcome in patients with very low-risk PCa and determine the predictable factors, including mpMRI findings, that affect the oncologic outcome. We believe that the findings of the study will play a significant role in establishing treatment guidelines in very low-risk PCa.

## 2. Materials and Methods

### 2.1. Study Design and Patients

We retrospectively retrieved the clinical and pathological data of individuals with very low-risk PCa who underwent mpMRI before RP at our institution between January 2010 and December 2016. Among them, patients who had undergone androgen deprivation therapy (ADT) before RP were excluded from the study. Before undergoing RP, all patients were diagnosed with PCa through a transrectal ultrasound (TRUS)-guided 12-core systematic needle biopsy. According to the NCCN guidelines, we defined patients with very low-risk PCa as those with cT1c, biopsy GS 6, PSA < 10 ng/mL, presence of disease in fewer than 3 biopsy cores, < 50% PCa involvement in any core, and PSA density < 0.15 ng/mL^2^ [9]. Additionally, we reviewed only patients with mpMRI based on the PI-RADS version 2 (PIRADS^v2^) [14], including standardized criteria for Likert scoring of multiparametric sequences (T1-weighted (T1W) and T2-weighted (T2W) imaging, diffusion-weighted imaging (DWI), apparent-diffusion coefficient (ADC), and dynamic contrast enhanced imaging (DCE)) using a 3.0T MRI system (Intera Achieva 3.0T, Phillips Medical System, Best, The Netherlands) [16]. All images were retrospectively reviewed by three experienced uroradiologists, who were blinded to the biopsy results and who conducted a consensus review of the mpMRI images of all patients. The suspicious lesions in the mpMRI were graded using a scoring system established by the PIRADS^v2^. Negative MRI findings were defined as having no grade 3 or higher region of interests without limitations on interpretation from mpMRI by radiologists [17,18]. Clinical characteristics of these patients, including age, body mass index (BMI), preoperative PSA, prostate volume measured by TRUS, Gleason score (GS) following prostate biopsy, and pathologic characteristics of specimens following RP were obtained by review of medical records at our institution. All pathologic diagnosis was performed by expert pathologists. Biopsy specimens obtained from centers other than our hospital were reviewed by our pathologists. Finally, the TNM stage was determined according to the 8th edition of the American Joint Committee on Cancer TNM staging system.

### 2.2. Follow-Up 

Postoperative PSA follow-up was performed monthly for the first 6 months, every 3 months for the second year, and every 6 months thereafter. BCR was defined as two consecutive increases in serum PSA ≥ 0.2 ng/mL following RP [19]. BCR-free survival was defined as the time from RP to BCR. The follow-up period was calculated from the time of RP to the date of the last known contact with the patient. 

### 2.3. Statistical Analysis

Univariate and multivariate Cox proportional hazards regression analyses were performed to assess the association between the baseline parameters and BCR-free survival. In addition, univariate and multivariate logistic regression were carried out to analyze the significant factors for the pathologic GS upgrade. Significant variables from univariate analysis were included in the multivariate analysis. In addition, Kaplan–Meier analysis with log-rank tests was performed to estimate and compare the oncologic outcomes according to the pathologic GS upgrade. *p* < 0.05 was considered statistically significant. All statistical analysis was performed using the SPSS Statistics software, version 23.0 (IBM, Armonk, NY, USA). 

### 2.4. Ethical Statement

This study was approved by the Institutional Review Board of the Severance Hospital (4-2018-1204).

## 3. Results

### 3.1. Patient and Disease Characteristics

A total of 253 patients (all Asian men), defined as very low-risk PCa according to the NCCN guidelines, were included in the study. Their baseline clinical and pathological features are presented in Table 1. The median age and BMI for patients was 65 years and 24.6 kg/m^2^ (interquartile range (IQR) 59–69, 22.4–26.2), respectively. The median prostate volume, as measured by TRUS, was 40.9 mL (IQR 34.1–50.0), the median PSA level was 4.6 ng/mL (IQR 3.9–5.6), and the median PSA density was 0.12ng/mL^2^ (IQR 0.09–0.14). The median interval period between prostate biopsy and RP was 29 days (IQR 14–48). The median follow-up from RP was 53 months (IQR 33–74). In addition, 163 patients presented with suspicious lesion and 90 with no lesion on mpMRI. In specimens following RP, 161 cases were GS 6 (63.6%) and 85 were GS 7 (33.6%). Of these, ISUP 2 was 72 (28.5%) and ISUP 3 was 13 (5.1%). Furthermore, GS above 8 was reported in seven cases (2.8%). Therefore, the pathologic GS upgrade rate over ISUP 2 was 92/253 (36.4%). Extracapsular extension (ECE) was present in 55 cases (21.7%), and positive surgical margins (PSM) were involved in 55 cases (21.7%). Seminal vesicle invasion was not observed in any of the patients. Perineural invasion (PNI) was reported in 87 cases (34.4%), lymphovascular invasion (LVI) in one case (0.4%), and high-grade prostatic intraepithelial neoplasia (HGPIN) in 99 cases (39.1%). Furthermore, there were no distant metastases and cancer-specific deaths during the observation period. 

### 3.2. Oncologic Outcomes

During the follow-up period we observed 26 cases (10.3%) with BCR. Univariable and multivariable Cox regression analyses were performed with each clinical parameter for BCR. In these analyses, the only factor associated with BCR in very low-risk patients was the increase in the pathologic GS (Hazard ratio (HR): 2.185, 95% confidence interval (CI): 1.010–4.728, *p*-value 0.048). The PSA and PSA density already defined by the NCCN guidelines in very low-risk patients were not statistically significant (*p* values 0.169 and 0.335, respectively) for BCR. Age, prostate volume, and suspicious lesion on mpMRI were also statistically insignificant (*p*-value: 0.110, 0.711, and 0.517, respectively). In addition, the pathologic findings through RP, such as ECE, PSM, LVI, PNI, and HGPIN were not statistically significant related to BCR (*p*-value: 0.638, 0.332, 0.799, 0.854 and 0.137, respectively) (Table 2). We also analyzed the BCR and the pathologic GS upgrade relationship through the Kaplan–Meier curves. This showed that BCR-free survival was statistically significant according to the pathologic GS upgrade (log-rank test, *p*-value: 0.042) (Figure 1).

### 3.3. Preoperative Predictors in Relation to the Pathologic Gleason Score Upgrade

In this study, we used univariate and multivariate logistic regression analyses to identify predictors associated with the pathologic GS upgrade. In these analyses, PSA (Odds ratio (OR): 1.353; 95% CI: 1.134–1.616; *p*-value: 0.010), PSA density (OR: 1.160; 95% CI: 1.053–1.279; *p*-value: 0.013), and suspicious lesion on mpMRI (OR: 1.995; 95% CI: 1.105–3.603; *p*-value; 0.019) constituted independent preoperative predictors of the presence of the pathologic GS upgrade at RP in both univariate and multivariate models (Table 3). In the multivariate regression equation, the r squared value of Nagelkerke was 0.137. It was able to explain the pathologic GS upgrade phenomenon by 13.7%. Among 163 patients with suspicious lesions, 93(57.1%) had GS upgrade. Twenty-three (34.3%) had GS upgrade among 67 patients with invisible lesion. In our study, diagnosis of the pathologic GS upgrade with mpMRI revealed sensitivity of 42.9%, specificity of 74.4%, positive predictive value of 75.3%, and negative predictive value of 41.9%. In addition, presence of suspicious lesion on mpMRI was 2.2 times more likely to have pathologic GS upgrade than invisible lesion.

## 4. Discussion

With the increasing number of patients with PCa, the classification of very low-risk PCa by several guidelines is based on the basic principle that overtreatment should not cause harm to the patient. Even an experienced urologist cannot deny that after RP, incontinence and impotence may occur in some patients [20]. Therefore, we are also trying to avoid overtreatment through active surveillance in patients with very low-risk PCa [21]. The prognosis for the currently defined very low-risk PCa patients is generally good, but as mentioned earlier, poor oncologic outcomes exist in some patients [11]. Therefore, we believe that if we can predict oncologic outcome through preoperative evaluation in very low-risk PCa patients, the early selection of active treatments, such as RP, may provide greater benefit to them. In some guidelines, mpMRI is not an indication in very low-risk PCa patients. However, recently, the development and standardization of mpMRI (through PI-RADS) have played a significant role in the diagnosis and treatment decisions of localized PCa. Since 2010, our institution has been conducting most of the mpMRI, including DWI and ADC with pre-evaluation in PCa patients. Thus, we were able to analyze mpMRI in a sufficient number of very low-risk PCa patients. Hence, we analyzed the oncologic outcome in patients with very low-risk PCa and investigated predictors, including mpMRI. The prospective LAPPRO trial also suggests that mpMRI may be helpful in very low-risk Pca [11]. Analysis of the oncologic outcome in these patients revealed adverse pathologic features and BCR in follow-up periods in some patients. In particular, the pathologic GS upgrade was the only independent prognostic factor associated with BCR. Discordance between biopsy GS and pathologic GS has already been reported in many articles [22,23]. It is known to occur because of the multifocal nature of PCa [24]. This discordance is reduced through continuous revision of ISUP [25,26], but it is difficult to completely prevent it. Several papers have been published on the relationship between mpMRI and GS [27,28,29]. We investigated independent factors that could predict the GS upgrade in patients with very low-risk PCa with reference to these studies. In our study, PSA, PSA density, and suspicious lesions of mpMRI were statistically significant factors for GS upgrade. In very low-risk patients, PSA and PSA density are controlled by definition. Even within the controlled limits PSA and PSA density were statistically significant. As already known, they may be predictors of the powerful oncologic outcome in PCa [16,30]. In addition, the suspicious lesions on mpMRI were also related to the GS upgrade. This suggests that mpMRI can play an important role as a pre-evaluation diagnostic tool in very low-risk PCa. We suggest that we should consider the treatment modality of very low-risk PCa patients by combining mpMRI findings, PSA and PSA density. Therefore, we believe that mpMRI, the most widely used image tool in PCa, should be used aggressively in patients with very low-risk PCa. However, our study had several limitations. First, this was a retrospective review of data from patients treated at a single institution. Therefore, PI-RADS scoring for suspicious lesions could not be performed accurately in all cases. The experienced uroradiologist reanalyzed the images, but it was determined to include bias in the PI-RADS scoring for suspicious lesions. Finally, we classified them as suspicious lesions (PIRADS ^v2^ 3–5) and no suspicious lesions (PIRADS ^v2^ 0–2). Therefore, adding the PI-RADS score in multicenter and prospective studies may aid in improving the accuracy of mpMRI in patients with very low PCa. Moreover, we did not perform MRI fusion biopsy in this study. In case involving PCa, performing MRI fusion biopsy may be useful, as suggested in the PROMIS study. It may reduce GS discordance between biopsy and surgery and reduce the prevalence of unnecessary examinations and surgery. We think that addition of MRI fusion target biopsy and addition analysis of lesions, such as periprostatic fat in MRI finding may be helpful in increasing the accuracy of the study [29,31]. Second, mpMRI scans were obtained after prostate biopsy. There may be some limitations on image interpretation, such as in case of hemorrhage, compared to prebiopsy images [32]. This might have caused the image quality to deteriorate. However, we consulted with experienced uroradiologist and confirmed that this did not limit interpretation of the image. Finally, in our study, BCR was observed in 26 patients during the follow up period. This may be a lack of numbers to analysis accurate statistics. Thus, in our study, the finding of mpMRI was related to the GS upgrade but not statistically significant with BCR. We think that increasing number of patients and follow-up periods through multicenter studies can obtain better results.

Despite these limitations, our study remains informative for clinicians who treat patients with very low-risk PCa. We investigated the long-term oncologic outcomes of very low-risk PCa with an established protocol. No patients were treated with adjuvant ADT or radiotherapy until BCR. This allowed us to observe the natural course of BCR after RP. Based on this, our study provides criteria for predicting adverse oncologic outcomes of patients with very low-risk PCa. We demonstrated the usefulness of mpMRI in very low PCa by identifying the relationship between GS upgrade and mpMRI.

We suggest that inclusion of mpMRI as a diagnostic indication in the currently proposed very low-risk PCa criteria will be of considerable benefit in clinical practice. This may be a criterion for clinicians to consider aggressive treatment in patients with very low-risk PCa. 

## 5. Conclusions

In patients with very low-risk PCa, the pathologic GS upgrade after RP is not uncommon, and this had a negative impact on the oncologic outcome in our study. The mpMRI based on PIRADS^v2^ performed in these patients is favorable to predict the pathologic GS upgrade. Currently, mpMRI is not an indication in very low-risk patients in the NCCN guidelines. However, we suggest that the implementation of mpMRI can be beneficial in determining the treatment strategy for these patients.

## Figures and Tables

**Figure 1 jcm-08-00542-f001:**
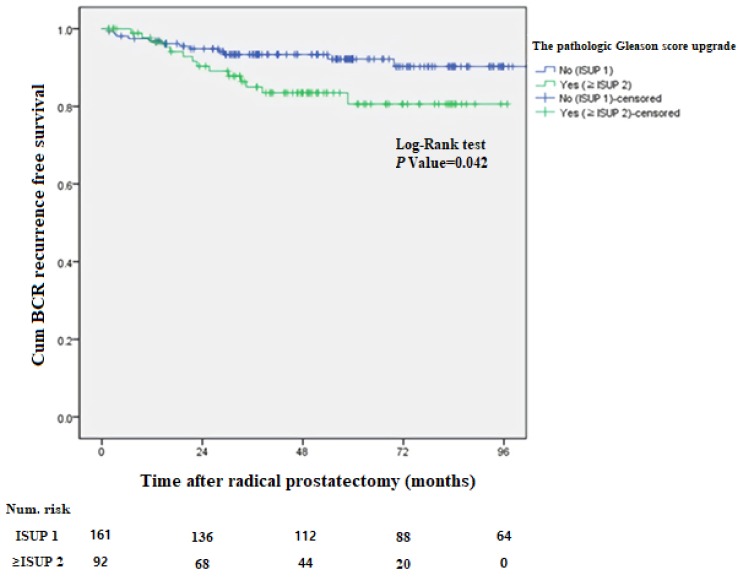
Kaplan–Meier curves for biochemical recurrence (BCR)-free survival in patients according to the pathologic Gleason score upgrade based on International Society of Urological Pathology (ISUP).

**Table 1 jcm-08-00542-t001:** Baseline patient characteristics.

	Median	IQR
Age, year	65	59–69
BMI, kg/m^2^	24.6	22.4–26.2
Prostate volume, mL	40.9	34.1–50.0
PSA level, ng/mL	4.6	3.9–5.6
PSA density, ng/mL^2^	0.12	0.09–0.14
MRI suspicious lesion	N	%
No	90	35.6
Yes	163	64.4
Follow up period after RP, months	53.0	33–74
Pathologic Gleason score	N	%
3 + 3 (ISUP 1)	161	63.6
3 + 4 (ISUP 2)	72	28.5
4 + 3 (ISUP 3)	13	5.1
≥8 (≥ISUP 4)	7	2.8
Pathologic T stage	N	%
<T2	198	78.3
≥T3	55	21.7
ECE	N	%
	55	21.7
SVI	N	%
	0	0
PSM	N	%
	55	21.7
LVI	N	%
	1	0.4
PNI	N	%
	87	34.4
HGPIN	N	%
	99	39.1
BCR	N	%
	26	10.3

IQR, interquartile range; BMI, body mass index; PSA, prostate-specific antigen; MRI, magnetic resonance imaging; RP, radical prostatectomy; ISUP, international society of urological pathology; ECE, extracapsular extension; SVI, seminal vesicle invasion; PSM, positive surgical margins; LVI, lymphovascular invasion; PNI, perineural invasion; HGPIN, Prostatic intraepithelial neoplasia, high grade; BCR, biochemical recurrence.

**Table 2 jcm-08-00542-t002:** Univariate and multivariate analyses of factors associated with biochemical recurrence.

Variable	Univariate		Multivariate	
	HR (95% CI)	*p*-Value	HR (95% CI)	*p*-Value
Age, year	1.050 (0.988–1.115)	0.110		
PSA, ng/mL	1.185 (0.930–1.509)	0.169		
BMI, kg/m^2^	0.990 (0.847–1.157)	0.903		
Prostate volume, mL	1.004 (0.983–1.025)	0.711		
PSA density, ng/mL^2^	0.941 (0.831–1.065)	0.335		
MRI suspicious lesion				
No	1 (Ref)			
Yes	1.317 (0.573–3.030)	0.517		
Pathologic Gleason upgrade				
No (ISUP 1)	1 (Ref)		1 (Ref)	
Yes (≥ISUP 2)	2.185 (1.010–4.728)	0.048	2.185 (1.010–4.728)	0.048
ECE				
No	1 (Ref)			
Yes	1.232 (0.517–2.935)	0.638		
PSM				
No	1 (Ref)			
Yes	1.511 (0.657–3.479)	0.332		
LVI				
No	1 (Ref)			
Yes	0.049 (0.000–Max)	0.799		
PNI				
No	1 (Ref)			
Yes	1.080 (0.477–2.443)	0.854		
HGPIN				
No	1 (Ref)			
Yes	1.815 (0.827–3.986)	0.137		

HR, hazard ratio; PSA, prostate-specific antigen; BMI, body mass index; MRI, magnetic resonance imaging; ECE, extracapsular extension; PSM, positive surgical margins; LVI, lymphovascular invasion; PNI, perineural invasion; HGPIN, Prostatic intraepithelial neoplasia, high grade.

**Table 3 jcm-08-00542-t003:** Univariate and multivariate analyses of factors associated with the pathologic Gleason score upgrade.

Variable	Univariate		Multivariate	
	OR (95% CI)	*p*-Value	OR (95% CI)	*p*-Value
Age, year	1.044 (0.004–1.085)	0.031	1.038 (0.995–1.082)	0.103
PSA, ng/mL	1.353 (1.134–1.616)	<0.001	1.280 (1.060–1.546)	0.010
BMI, kg/m^2^	0.949 (0.854–1.054)	0.329		
Prostate volume, mL	0.990 (0.965–1.015)	0.418		
PSA density, ng/mL^2^	1.163 (1.057–1.279)	0.002	1.160 (1.053–1.279)	0.013
Interval period between Biopsy and RP	0.998(0.993–1.003)	0.392		
Biopsy positive core, No				
1	1 (Ref)			
2	0.786 (0.440–1.403)	0.414		
3	1.446 (0.701–2.983)	0.318		
MRI suspicious lesion				
No	1 (Ref)		1 (Ref)	
YES	2.193 (1.245–3.862)	0.007	1.995 (1.105–3.603)	0.019

OR, odds ratio; CI, confidence interval; PSA, prostate-specific antigen; BMI, body mass index; RP, radical prostatectomy; MRI, magnetic resonance imaging.

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
