# Peer review of "Clinical Significance of Multiparametric Magnetic Resonance Imaging as a Preoperative Predictor of Oncologic Outcome in Very Low-Risk Prostate Cancer"

_jcm, 2019, doi:10.3390/jcm8040542_

Round 1
Reviewer 1 Report
1)introduction can be improved. It is now from PROTECT study results that radical prostatectomy and radical radiotherapy and active surveillance are equally good options after 10 year follow up. Authors mention that Radical prostatectomy remains the best treatment for low risk prostate cancer.
2)Randomised controlled trial of diagnostic accuracy of mpMRI and TRUS biopsies, (PROMIS study) results have been published in lancet 2017. the trial results contradict the retrospective review presented here significantly. PROMIS study reports sensitivity of MP-MRI for clinically significant cancer was 93% (95% CI 88–96%) and negative predictive value 89% (83–94%). Specificity of MP-MRI was 41% (36–46%) with positive predictive value 51% (46–56%).
In spite of the limitations described by the authors such as retrospective nature of the study and small numbers, the difference is significant. Explanation, why this may have been should be included in the discussion.
Author Response
Dear Reviewer
Thank you for your thoroughly reviewing our manuscript (jcm-479461) entitled “Clinical significance of multiparametric magnetic resonance imaging as a preoperative predictor of oncologic outcome in very low-risk prostate cancer” Also, we are grateful for the chance to revise our manuscript. Our manuscript has been carefully revised according to the reviewers’ comments. Please find our responses to the reviewer’ comments beginning on the next page.
We hope that our revised paper is acceptable for publication in Journal of Clinical Medicine, and we look forward to receiving your final decision.
Thanks, again.
Sincerely,
Comment
1) introduction can be improved. It is now from PROTECT study results that radical prostatectomy and radical radiotherapy and active surveillance are equally good options after 10 year follow up. Authors mention that Radical prostatectomy remains the best treatment for low risk prostate cancer.
Answer
Thank you for your comment. We have modified this paper according to your feedback. We think there was something wrong with the choice of terms. Sorry. We also believe that localized prostate cancer therapy should be carefully selected for surgery, radiotherapy, and active surveillance to provide appropriate treatment for patients. The sentence has been modified. Thank you. (Location of the content: 37-39)
Comment
2) Randomised controlled trial of diagnostic accuracy of mpMRI and TRUS biopsies, (PROMIS study) results have been published in lancet 2017. the trial results contradict the retrospective review presented here significantly. PROMIS study reports sensitivity of MP-MRI for clinically significant cancer was 93% (95% CI 88–96%) and negative predictive value 89% (83–94%). Specificity of MP-MRI was 41% (36–46%) with positive predictive value 51% (46–56%).
In spite of the limitations described by the authors such as retrospective nature of the study and small numbers, the difference is significant. Explanation, why this may have been should be included in the discussion.
Answer
Thank you for reviewing and comparing our research with the PROMIS study. We have checked up our paper according to your feedback.
In our study, diagnosis of the pathologic GS upgrade with mpMRI revealed sensitivity of 42.9%, specificity of 74.4%, positive predictive value of 75.3%, and negative predictive value of 41.9%. (Location of the content: 150-152)
<< This describes the sensitivity, specificity, PPV, and NPV for MRI of patients who have been upgraded to more than GS7 (3 + 4) after RP among very low risk patients enrolled in our study.
The PROMIS study reported the results for clinical significant cancer of prostate biopsy. Therefore, our content may differ from the PROMIS study figures mentioned above.
In our previous study, 12.8% (242/1895) of patients following radical prostatectomy with prostate cancer (excluded neoadjuvant therapy or incomplete mpMRI) had invisible lesion in mpMRI. This was similar to previously reported studies included the PROMIS study.
(Reference) Chung, D. Y .; Koh, D. H .; Goh, H.J .; Kim, M.S .; Lee, J. S .; Jang, W.S .; Choi, Y.D. Clinical significance and predictors of oncologic outcome after radical prostatectomy for invisible prostate cancer on multiparametric mri. Bmc Cancer 2018, 18.
All patients with very low prostate cancer risk in this study were diagnosed with TRUS guided systematic biopsy. Very low risk prostate cancer patients are clinically insignificant prostate cancer patients based on PROMIS study. However, in our study, there was a patient who eventually had significant prostate cancer among these patients.
We did not perform MRI fusion biopsy in this study. We think this was one of limitations in our study. In prostate cancer, performing MRI fusion biopsy is a good method as suggested in the PROMIS study. It may be to reduce GS Discordance between Biopsy and surgery and to reduce unnecessary examinations and surgery. Therefore, we will add this to the next study.
Thank you for reminding us of good research once again, and we believe that your comments has increased the value of our research. Thank you. (Location of the content: 206-211)
Reviewer 2 Report
This article demonstrates that multiparametric MRI can be used to predict Gleason Score upgrade in their patient population and thereby help predict biochemical recurrence. The ability to predict whether a patient will experience BCR is important because this could help prevent overtreatment. A PubMed search revealed that multiparametic MRI has been used in multiple other prostate cancer studies, including one in which the authors show that it can be used to predict upgrading (https://www.ncbi.nlm.nih.gov/pubmed/30895901). The authors spend more time reviewing this literature (both in the Introduction and Discussion sections), and make note of what their study adds to our understanding of the utility of this methodology.
Some sections of the manuscript are well written, while others are poorly written and hard to follow. I recommend that editing services are used.
1. The ‘RP’ acronym should be put it brackets following first mention of radical prostatectomy in the abstract
2. Line 26 (and throughout the article): I think it would be more appropriate to state ‘prognostic tool’ in place of ‘diagnostic tool’, as these are patients who have already been diagnosed with prostate cancer (mpMRI would be used as a prognostic tool).
3. Line 26 and 42 (and throughout the article): How is oncologic outcome defined? BCR? If yes, then simply state BCR. If not, then please define.
4. The sentence on line 36/37 should be reconsidered; there are other treatment modalities available and whether RP is the ‘best’ treatment is debatable/is context dependent
5. Methods section: What was the median time between diagnosis and RP? Please comment on whether the length of time between diagnosis and RP could have impacted GS upgrade.
6. Figure 1: This figure is ‘fuzzy’/out of focus – this needs to be fixed. The x-axis should state ‘time after radical prostatectomy (months)’, the y-axis should stable ‘Cum BCR recurrence free survival’.
7. Table 1 (patient characteristics) should be included prior to figure 1. Patient race should also be included in this table or at least noted at the beginning of the results section (were the majority of these patients Asian?).
8. Please state the r squared values for the regression analyses and comment on these.
Author Response
Dear Reviewer
Thank you for your thoroughly reviewing our manuscript (jcm-479461) entitled “Clinical significance of multiparametric magnetic resonance imaging as a preoperative predictor of oncologic outcome in very low-risk prostate cancer” Also, we are grateful for the chance to revise our manuscript. Our manuscript has been carefully revised according to the reviewers’ comments. Please find our responses to the reviewer’ comments beginning on the next page.
We hope that our revised paper is acceptable for publication in Journal of Clinical Medicine, and we look forward to receiving your final decision.
Thanks, again.
Sincerely,
Comment
This article demonstrates that multiparametric MRI can be used to predict Gleason Score upgrade in their patient population and thereby help predict biochemical recurrence. The ability to predict whether a patient will experience BCR is important because this could help prevent overtreatment. A PubMed search revealed that multiparametic MRI has been used in multiple other prostate cancer studies, including one in which the authors show that it can be used to predict upgrading (https://www.ncbi.nlm.nih.gov/pubmed/30895901). The authors spend more time reviewing this literature (both in the Introduction and Discussion sections), and make note of what their study adds to our understanding of the utility of this methodology.
Answer
Thank you for your comment. We have modified this paper according to your feedback. Due to your comment, we think that this paper has become a more valuable literature.
Thank you for introducing good research. We read the paper well. In our study, there was no statistical significance with BMI and Gleason upgrade, but the research gave us the importance of periprostatic fat. Thank you. We have added this study as a reference. We will be able to expect good results from further consideration of this information in our next study. Thank you. (Location of the content: 206-211)
Comment
The ‘RP’ acronym should be put it brackets following first mention of radical prostatectomy in the abstract :
Answer
Thank you for your comment. We have modified this contents according to your feedback.
(Location of the content: 20)
Comment
Line 26 (and throughout the article): I think it would be more appropriate to state ‘prognostic tool’ in place of ‘diagnostic tool’, as these are patients who have already been diagnosed with prostate cancer (mpMRI would be used as a prognostic tool).
Answer
Thank you for your careful comment. We have modified this contents according to your feedback.
(Location of the content: 27)
Comment
Line 26 and 42 (and throughout the article): How is oncologic outcome defined? BCR? If yes, then simply state BCR. If not, then please define.
Answer
Thank you for your comment. We have modified this contents according to your feedback.
(Location of the content: 27 / 43-46)
Comment
The sentence on line 36/37 should be reconsidered; there are other treatment modalities available and whether RP is the ‘best’ treatment is debatable/is context dependent
Answer
Thank you for your comment. We have modified this paper according to your feedback. We think there was something wrong with the choice of terms. Sorry. We also believe that localized prostate cancer therapy should be carefully selected for surgery, radiotherapy, and active surveillance to provide appropriate treatment for patients. The sentence has been modified. Thank you.
(Location of the content: 37-39)
Comment
Methods section: What was the median time between diagnosis and RP? Please comment on whether the length of time between diagnosis and RP could have impacted GS upgrade.
Answer
Thank you for your comment. We have modified this paper according to your comment.
The median interval period between prostate biopsy and RP was 29 days (IQR 14-48)
The Interval period and GS upgrade were not statistically significant. (Location of the content: 109-110)
We have added this to Table 3.
Interval period between Biopsy and RP | 0.998(0.993-1.003) | 0.392 |
Comment
Figure 1: This figure is ‘fuzzy’/out of focus – this needs to be fixed. The x-axis should state ‘time after radical prostatectomy (months)’, the y-axis should stable ‘Cum BCR recurrence free survival’
Answer
Thank you for your comment. We have modified this paper according to your comment. We modified the terms and adjusted the image quality and added number at risk
Comment
Table 1 (patient characteristics) should be included prior to figure 1. Patient race should also be included in this table or at least noted at the beginning of the results section (were the majority of these patients Asian?).
Answer
Thank you for your comment. We have modified this paper according to your comment. We have added this contents.
All patients included in this study were Asian men. (Location of the content: 104)
Comment
Please state the r squared values for the regression analyses and comment on these.
Answer
Thank you for your comment. We have modified this paper according to your comment. We have added this contents.
In the multivariate regression equation, the r squared value of Nagelkerke was 0.137. It was able to explain the pathologic GS upgrade phenomenon by 13.7%. (Location of the content: 147-148)
Round 2
Reviewer 2 Report
My concerns have been addressed; thank you